# Application of a Cold Dry Air Provocation Test in Pediatric Patients with Asthma

**DOI:** 10.3390/children9060920

**Published:** 2022-06-19

**Authors:** Ji Young Ahn, Bong Seok Choi

**Affiliations:** 1Department of Pediatrics, College of Medicine, Yeungnam University, Daegu 42415, Korea; jy4413@gmail.com; 2Department of Pediatrics, School of Medicine, Kyungpook National University, Daegu 41944, Korea

**Keywords:** asthma, cold dry air provocation, chronic inflammatory airway disease, child

## Abstract

Asthma is a chronic inflammatory airway disease characterized by reversible airway obstruction and airway hyperreactivity. We proposed a cold dry air (CDA) provocation test and investigated its application in pediatric patients with asthma. We enrolled 72 children and adolescents older than 5 years who presented to our hospital with chronic cough, shortness of breath, and wheezing. We analyzed the results of allergy, pulmonary function, methacholine provocation, and CDA provocation tests. The FEV1 change 5 min after the provocation was recorded as CDA5 dFEV1; that after 15 min was recorded as CDA15 dFEV1. PT10 was the provocation time causing a 10% decrease in FEV1; a decrease of >10% in dFEV1 was considered a positive CDA test. Among the 72 subjects, 51 were diagnosed with asthma. A positive CDA test in patients with asthma correlated with non-eosinophilic asthma. In patients with asthma, sputum eosinophils and eosinophil cationic protein (ECP) levels of the patients with a positive CDA test were significantly lower than those of patients with a negative test. CDA5 dFEV1 correlated with PC20 and total immunoglobulin E. CDA15 dFEV1 correlated with PC20, sputum eosinophils, and ECP. PT10 became shorter as the peripheral blood eosinophil, FVC, FEV1, FEV1/FVC, and FEF25-75 decreased. The CDA provocation test showed airway hyperreactivity to non-specific stimuli, a high correlation with non-eosinophilic asthma, and the possibility of assessing asthma severity via PT10.

## 1. Introduction

Asthma is a chronic airway inflammatory disease with a diverse phenotype characterized by reversible airway obstruction and airway hyperresponsiveness [1,2]. The major diagnostic tests for asthma include spirometry and the bronchial provocation test [3,4].

There are specific and non-specific tests for bronchial provocation, in which the methacholine provocation test is representative of a non-specific test. Other non-specific bronchial provocation tests include tests using histamine, mannitol, exercise, and hyperventilation [5,6]. Non-specific bronchial provocation tests are divided into direct and indirect methods, according to the mechanism. Direct bronchial provocation tests directly stimulate airway smooth muscles using methacholine or histamine to cause airway constriction, and indirect bronchial provocation tests constrict the airway through changes of osmotic pressure in the airway with mannitol, exercise, and hyperventilation. Direct bronchial provocation tests have high sensitivity and low specificity and help diagnose asthma, while indirect bronchial provocation tests have low sensitivity and high specificity and are useful for the confirmation of asthma. Both tests have increased sensitivity and specificity; however, they cannot be performed simultaneously on the same day. Therefore, direct bronchial provocation tests, such as the methacholine provocation test, are mainly used in clinical practice [7].

An exercise-induced test, which is an indirect bronchial provocation test, requires equipment and may be difficult to perform in children. The hypertonic saline provocation test, which is another indirect bronchial provocation test, requires the patient to inhale hypertonic saline through a nebulizer after a complicated preparation process and can cause asthma attacks in patients with severe asthma.

We proposed a cold dry air (CDA) provocation test as an indirect bronchial provocation that can be easily performed in children who can perform pulmonary function tests. The CDA provocation test is not considered a diagnostic test for asthma such as the methacholine provocation test but aims to identify patients who have bronchoconstriction to CDA in some phenotypes of asthma. This study aimed to investigate the clinical implication of the CDA provocation test in children with asthma.

## 2. Materials and Methods

### 2.1. Subjects

We enrolled 72 children and adolescents older than 5 years of age who visited Kyungpook National University Children’s Hospital between August 2018 and December 2020 for evaluation of chronic cough, shortness of breath, or wheezing. We excluded subjects who had signs of acute infections, such as fever, or abnormal findings on chest radiography. Subjects underwent the ImmunoCAP (UniCAP; Pharmacia, Uppsala, Sweden) or skin prick test as allergy tests and performed the methacholine provocation test and CDA provocation test with an interval of more than one day. The study was approved by the Institutional Review Board of Kyungpook National University Chilgok Hospital (No. 2021-05-009) on 21 May 2021.

Asthma was diagnosed when the provocative concentration of methacholine caused a 20% decrease in forced expiratory volume in 1 s (FEV1, PC20) in the methacholine provocation test at 16 mg/mL or less, while the subjects had symptoms such as cough, wheezing, or shortness of breath [8,9]. Non-asthmatic subjects were defined as “other.” Among asthma patients, those with typical asthma symptoms, such as cough, wheezing, and shortness of breath, were referred to as having classic asthma, and those with only cough symptoms were referred to as having cough-variant asthma [10]. Allergic rhinitis was diagnosed when one or more inhalant allergen was positive in the ImmunoCAP and/or skin prick test, with rhinitis symptoms, such as runny nose, stuffy nose, sneezing, itchy nose, and itchy eyes. Non-allergic rhinitis was diagnosed when the inhalant allergen was negative and rhinitis symptoms were present [11,12]. Among asthma patients, those with allergic and/or non-allergic rhinitis were classified into asthma with rhinitis group, while those who had asthma only were classified into the asthma-only group.

### 2.2. Methods

#### 2.2.1. Blood Tests

The serum total eosinophil count was measured using an automated hemocytometer, while serum total immunoglobulin E (IgE) and specific IgE levels were measured using the CAP radioallergosorbent technique. Six allergens were tested: *Dermatophagoides farinae*, *Dermatophagoides pteronyssinus*, dog dander, cat dander, *Alternaria*, and *Aspergillus fumigatus*. Values of 0.35 ku/L or higher were defined as positive. Those who had positive results for any allergens on either specific IgE or skin prick test were defined as atopic, while those who had negative results were defined as non-atopic.

#### 2.2.2. Skin Prick Tests

Subjects taking antihistamines or oral steroids discontinued the medication three days before the test. Skin prick tests (Bencard Allergie GmbH, Munich, Germany) were conducted on 18 types of allergens, including birch, alder, hazel, pine, oak, Bermuda, timothy, orchard, ragweed, mugwort, *D. farinae*, *D. pteronyssinus*, *Acarus siro*, *Tyrophagus*, dog dander, cat dander, *Alternaria*, and *Aspergillus* [13]. Histamine and normal saline were used as positive and negative controls, respectively, and the size of the wheal was measured after 15 min. A positive result was defined as a wheal of 3 mm or greater than that of the histamine control.

#### 2.2.3. Induced Sputum Tests

Induced sputum tests were performed according to the guidelines of the Korean Academy of Asthma, Allergy, and Clinical Immunology [6]. After inhaling nebulized sterile 3% hypertonic saline solution for 5 min through an ultrasonic nebulizer (OMRON NE-U17; OMRON Matsusaka Co., Ltd., Kyoto, Japan), the sputum was expectorated into a plastic tube. The subjects repeated this procedure four times. After the volume of induced sputum was determined, quadrupled volumes of 0.1% dithiothreitol were added. The samples were mixed in a vortex mixer for 30 s. After 15 min, an equal volume of saline was added and mixed in a vortex mixer for 15 s. Induced sputum was placed in a 1 mL centrifuge tube and centrifuged at 2000 rpm for 5 min. The upper layer of the fluid was separated and placed in a 5 mL tube on ice. The cell separation was resuspended in normal saline and centrifuged at 450 rpm for 6 min to prepare the slides. After Wright staining, the cell fraction of induced sputum was calculated as a percentage. Among asthma patients, eosinophilic asthma was classified when the eosinophil fraction was 3% or higher in the induced sputum, and non-eosinophilic asthma when the eosinophil fraction was less than 3%.

#### 2.2.4. Pulmonary Function Tests and Spirometry

Using a spirometer (Vmax 20; Viasys, San Diego, CA, USA), we measured forced vital capacity (FVC), FEV_1_, and FEV_1_/FVC according to American Thoracic Society (ATS) guidelines [9]. The predicted lung capacity was calculated based on data from the Third National Health and Nutrition Examination Survey [14]. Subjects using inhaled bronchodilators discontinued the medication at least 48 h before the test.

#### 2.2.5. Methacholine Provocation Tests

We conducted a methacholine provocation test using methacholine chloride (provocholine; Methapharm Inc., Brantford, ON, Canada) and measured FVC, FEV_1_, and PC20. The methacholine provocation test was based on the five-breath technique according to ATS guidelines and a dosing schedule using methacholine concentrations of 0.0625, 0.25, 1, 4, 16, and 25 mg/mL [8]. The subjects stopped treatment with a short-acting bronchodilator for 8 h, long-acting bronchodilator and theophylline for 48 h, antihistamine for 72 h, and leukotriene antagonist for 24 h. A positive methacholine provocation test was defined as a PC20 of 16 mg/mL or less.

#### 2.2.6. CDA Provocation Test

The subjects waited in the laboratory for 30 min to adapt to the temperature and humidity before the CDA provocation test. After 30 min, a baseline pulmonary function test was conducted, and the CDA provocation test was performed when the baseline FEV1 was over 70%. Hyperventilation was conducted with CDA (0 °C, humidity < 10%) through a face mask at a rate of 1 L/min/kg (25 × FEV1 per minute) for 4 min, and FEV1 was measured 5 min and 15 min after the test. If the decrease in FEV1 15 min after the CDA provocation test did not return to the baseline, a bronchodilator was inhaled, and a lung function test was conducted 10 min later [15,16,17]. The FEV1 change 5 min after the provocation was recorded as CDA5 dFEV1, and the FEV1 change after 15 min was recorded as CDA15 dFEV1. PT10 (provocation time causing a 10% decrease in FEV1) was defined as the time when FEV1 fell by greater than 10%, and a decrease of 10% or greater in dFEV1 was defined as the positivity of the CDA test [18,19].

### 2.3. Statistical Analysis

All statistical analyses were performed using PASW Statistics ver. 18.0 (SPSS Inc., Chicago, IL, USA). Quantitative values are expressed as means and standard deviations or as medians from minimum to maximum. Comparisons between both groups were performed using Student’s *t*-test and ANOVA analysis. Post hoc analysis was performed using Bonferroni analysis. In the correlation analysis, Pearson correlation was used for parametric variables, and Spearman correlation was used for non-parametric variables. Statistical significance was defined as *p* < 0.05.

We analyzed the characteristics of subjects, including sex, age, height, weight, and body mass index (BMI). We also analyzed the results of blood tests, including eosinophil counts, eosinophil cationic protein (ECP), total IgE, specific IgE, skin prick test, induced sputum fraction, pulmonary function test, methacholine provocation test, and CDA provocation test.

## 3. Results

The total number of subjects was 72, of whom 47 were male and 25 were female. The average age of the subjects was 9.1 years. The numbers of symptoms including cough, dyspnea, coughing during exercise, dyspnea during exercise, and cough when exposed to cold air are summarized in Table 1. We investigated the diagnoses of allergic rhinitis, atopic dermatitis, food allergy of the subjects prior to visiting our clinic, and family history of allergic diseases, including asthma. Among all subjects, 58 (80.6%) were atopic, and 14 (19.4%) were non-atopic.

Of all subjects, 51 (70.8%) had asthma, and 21 (29.2%) had other diseases, such as allergic rhinitis and non-allergic rhinitis. Among the asthma patients, 14 (27.5%) had eosinophilic asthma, and 37 (72.5%) had non-eosinophilic asthma. According to asthma symptoms, 37 (72.5%) had classic asthma, and 14 (27.5%) had cough-variant asthma. All subjects were classified as asthma or other diseases group, and their laboratory test results are shown in Table 2. The sputum eosinophil (%) (*p* = 0.014) and ECP (*p* = 0.001) were significantly higher in patients with asthma than in those with other diseases, and FEV1/FVC (*p* < 0.001) and PC20 (*p* < 0.001) were significantly lower.

### 3.1. CDA Provocation Tests in the Children with Suspected Asthma and/or Respiratory Allergies

In all subjects, no significant correlation was observed between age, height, weight, BMI, and positivity of the CDA provocation test. There was no significant correlation between positivity for CDA provocation and cough, dyspnea, dyspnea on exercise, response to cold air, history of atopic dermatitis, history of food allergy, frequency of asthma exacerbation for three years, exposure to smoking, family history of asthma, and family history of allergic diseases. There was no correlation between the positivity of CDA provocation and the results of spirometry and methacholine provocation test in all subjects. Sputum eosinophil (%) of the subjects who were positive for CDA provocation was significantly lower than that of those who were negative for CDA provocation (*p* = 0.020). No correlation was observed between positivity of CDA provocation and eosinophil counts, total IgE, ECP, atopy, the positivity of specific IgE, and positivity and positive numbers in the skin prick test. The results are shown in Table 3. CDA15 dFEV1 correlated with PC20 (r = 0.336, *p* = 0.017), and PT10 became shorter as FEV1/FVC (r = 0.547, *p* = 0.043) and FEF25-75 (r = 0.534, *p* = 0.049) decreased in all subjects.

### 3.2. CDA Provocation Test in Asthma Patients

Among all asthma patients, 27 were positive for the CDA provocation test, of which two (14.3%) had eosinophilic asthma and 25 (67.6%) had non-eosinophilic asthma. In addition, according to symptoms, classic asthma was present in 21 (56.8%) and cough-variant asthma in six (42.9%). In non-eosinophilic asthma, the positive rate of the CDA provocation test was significantly higher (*p* = 0.001) (Figure 1). In all patients with asthma, sputum eosinophil (%) (*p* = 0.019) and ECP (*p* = 0.047) of the subjects who were positive for CDA provocation were significantly lower than those of the subjects who were negative for CDA provocation (Table 4). In all asthma patients, CDA5 dFEV1 correlated with PC20 (r = 0.295, *p* = 0.049) and total IgE (r = −0.302, *p* = 0.033). CDA15 dFEV1 correlated with PC20 (r = 0.36, *p* = 0.015), sputum eosinophil (%) (r = 0.305, *p* = 0.039), and ECP (r = 0.296, *p* = 0.041) (Figure 2). PT10 became shorter as the eosinophil count (r = 0.731, *p* = 0.011), FVC (r = 0.636, *p* = 0.019), FEV1 (r = 0.611, *p* = 0.027), FEV1/FVC (r = 0.599, *p* = 0.031), and FEF25-75 (r = 0.615, *p* = 0.025) decreased and shorter as the BMI increased (r = −0.608, *p* = 0.027).

### 3.3. Comparison between Asthma with Rhinitis and Asthma-Only Group

Among all asthma patients, 33 (64.7%) were in the asthma with rhinitis group and 18 (35.3%) in the asthma-only group. There were no significant differences between the two groups in terms of age, sputum eosinophil count, eosinophil count, ECP, total IgE, number of positive skin prick test results, FVC, FEV1, FEV1/FVC, and PC20 (Table 5). We investigated correlations between the CDA provocation test and medical history of subjects, pulmonary function test, and laboratory test results in the asthma with rhinitis and asthma-only group. Nineteen patients were positive for CDA provocation in the asthma with rhinitis group, and eight patients were positive for CDA provocation in the asthma-only group. There was no significant difference between the two groups in positivity for CDA provocation and PT10.

In the asthma with rhinitis group, the decrease in CDA5 dFEV1 was greater as total IgE increased (r = −0.392, *p* = 0.026). In addition, the decrease in CDA15 dFEV1 was greater as sputum eosinophil (%) (r = 0.394, *p* = 0.031), eosinophil count (r = 0.364, *p* = 0.040), and ECP (r = 0.373, *p* = 0.035) decreased. PT10 was shorter as the eosinophil count decreased (r = 0.724, *p* = 0.042). In the asthma-only group, the decrease in CDA5 dFEV1 was greater in the absence of cough during exercise (*p* = 0.044), and the decrease in CDA15 dFEV1 was greater in females (*p* = 0.002) and non-atopy (*p* = 0.011).

### 3.4. Comparison between Methacholine and CDA Provocation Tests

According to the results of the methacholine and CDA provocation tests, we classified the patients into four groups and compared each with a medical history of subjects, results of pulmonary function tests, and laboratory tests. Sputum eosinophils (%) were significantly lower in the positive CDA group than in the negative CDA group among the subjects who were positive for methacholine provocation (*p* = 0.012). No significant differences were observed between the four groups in terms of other medical histories and test results.

### 3.5. Others

Among 21 patients with other diseases, seven (33.3%) were positive for CDA, four of which were allergic rhinitis, and two complained of chronic cough and dyspnea on exercise, with normal results of spirometry and methacholine provocation test. The other patient complained of chest pain and dyspnea but improved without treatment. There was no significant correlation between positivity of CDA provocation and medical histories of subjects, atopy, results of allergy test, spirometry, and methacholine provocation test.

## 4. Discussion

In this study, the CDA provocation test was relatively fast and feasible, and no particular adverse events emerged. CDA provocation showed a high association with non-eosinophilic asthma and showed the possibility of assessing the severity of asthma through PT10. In total, 27 out of 51 asthma patients had a positive CDA provocation (52.9%), and the decrease in CDA15 dFEV1 was greater as PC20 was reduced in all subjects and all asthma patients, which is consistent with the study of Modl et al. [16].

The CDA provocation test showed a higher positive rate in non-eosinophilic asthma than in eosinophilic asthma. It revealed airway hyperreactivity in patients with non-eosinophilic asthma. In patients with asthma, the decrease in CDA15 dFEV1 was greater as sputum eosinophil (%), and ECP reduced, and PT10 was shorter as eosinophil count reduced. The association between CDA provocation and non-eosinophilic asthma has not been reported in previous studies.

In this study, the PT10 of all subjects and all asthma patients was shorter as FEV1, and FEV1/FVC decreased. Therefore, there is a possibility that the PT10 of the CDA provocation test could be used as an indicator of asthma severity, but additional large-scale studies are needed. In addition, as in the study by Steinbacher et al. [20], who performed CDA provocation as a test for small airway involvement in pediatric asthma, our study showed that PT10 shortened as FEF25-75 reduced, indicating peripheral airway inflammation. The results showed that the CDA provocation test could be used as a test for small airway involvement.

The CDA provocation test uses the principles of change in the osmotic pressure in the airway. When CDA is inhaled, the mucous membrane of the airway surface becomes dry, which leads to an increase in the osmotic pressure of the airway surface. As the osmotic pressure on the airway surface increases, water from the surrounding cells moves toward the airway lumen, and the subepithelial cells contract. Constriction of the subepithelial cells induces the secretion of airway constriction-related inflammatory mediators, which leads to airway constriction [21,22]. In general, CDA provocation is correlated with methacholine provocation, and it shows airway hyperreactivity more meaningfully than the test performed through inhalation of a bronchoconstrictor [23,24,25]. In addition, CDA more accurately reflects airway inflammation [16,17,26]. However, some studies reported no association between CDA provocation and methacholine provocation [27,28].

CDA can be used to diagnose exercise-induced bronchoconstriction [29,30,31]. However, in this study, no significant correlation was observed between CDA provocation and cough and dyspnea during exercise. In addition, in this study, an exercise provocation test was not performed; thus, the association between the two tests could not be confirmed.

The methacholine provocation test, which is commonly used in clinical practice, requires a standardized aerosol and conducts the test through several concentration changes, such as 0.0625, 0.25, 1, 4, 16, and 25 mg/mL; therefore, sufficient time is required. In addition, the deposition pattern of the aerosol may vary depending on the compliance of the subjects, which may affect the results of the test [32]. The methacholine provocation test has high sensitivity but low specificity; thus, it can be positive in other lung diseases and rhinitis, as well as in some normal people [33]. The methacholine provocation test could be a false negative because methacholine can affect intrabronchial delivery depending on the storage, dilution, and amount of spray. In addition, the methacholine provocation test may be negative in pediatric patients with a short disease period, exercise-induced asthma, and non-atopic asthma [7,34,35,36]. In contrast, the CDA provocation test uses naturally occurring stimuli and has the advantage of shorter test time compared to the methacholine provocation test. In addition, problems caused by aerosol standardization or compliance of subjects can be avoided. Although the CDA provocation test has lower sensitivity than the methacholine provocation test, it has a higher specificity [37]. However, the CDA provocation test has a disadvantage in that an interval of more than 24 h is required between the two tests when performed simultaneously.

The limitations of this study are as follows. First, because of the small sample size, additional large-scale studies are needed. Second, an exercise-induced test was not performed; thus, a comparison thereof is impossible. Third, because there were patients who had omitted medical records related to symptoms during normal exposure to cold air and dyspnea on exercise, the evaluation of the correlation with CDA provocation might not have been sufficient. Therefore, additional studies on the association between the CDA provocation test results and the symptoms that appear when exposed to CDA are needed after developing an objective index of symptoms of CDA. Fourth, because the diagnostic criteria for asthma were defined as PC20, which is the positive criterion for the methacholine provocation test, at 16 mg/mL or less, there is a possibility that methacholine-negative asthma was lost. Fifth, the CDA provocation test was conducted using a positive pressure mask as used in previous studies, and it is possible that the effect of CDA might be reduced because a warming and humidifying effect occurred in the air inhaled through the nose. In a future study, performing the CDA provocation test using a mouthpiece can help obtain more distinct results.

In conclusion, this study revealed that the CDA provocation test in pediatric asthma patients showed airway hyperreactivity to non-specific stimuli, was highly correlated with non-eosinophilic asthma, and demonstrated the potential of PT10 for assessing the severity of asthma. In addition, asthma management may be performed more effectively by identifying patients sensitive to CDA and minimizing exposure to these situations through the CDA provocation test. Additional large-scale studies compensating for the limitations are needed.

## Figures and Tables

**Figure 1 children-09-00920-f001:**
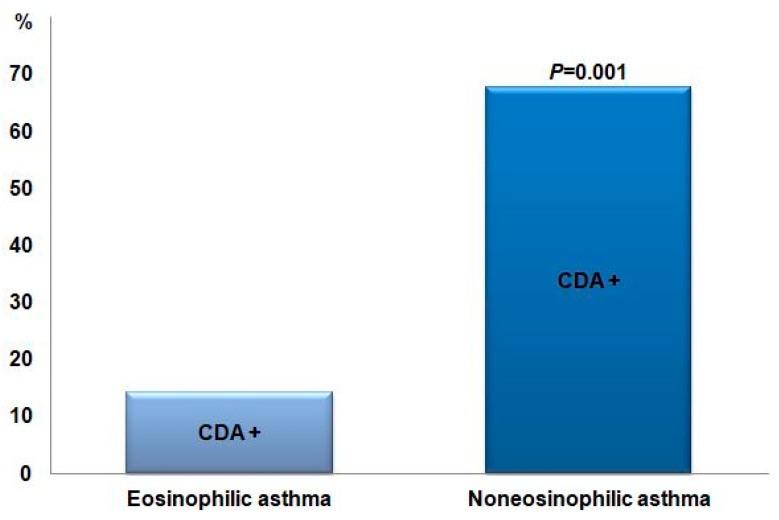
Numbers of positive CDA provocation tests in asthma patients. CDA—cold dry air.

**Figure 2 children-09-00920-f002:**
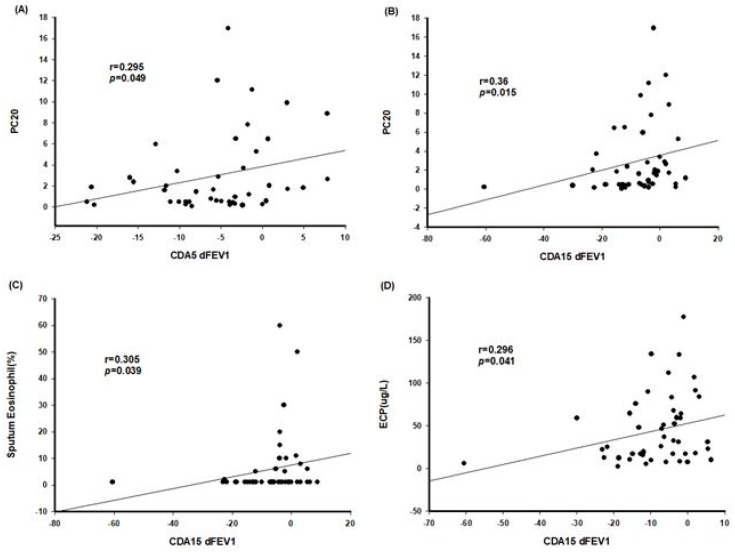
Correlation between CDA results and other factors in asthma patients. CDA—cold dry air; PC20—provocative concentration of methacholine causing a 20% decrease in FEV1; dFEV1—change in forced expiratory volume in 1 s; ECP—eosinophilic cationic protein. (**A**) CDA5 dFEV1 and PC20 (**B**) CDA15 dFEV1 and PC20 (**C**) CDA15 dFEV1 and Sputum Eosinophil(%) (**D**) CDA15 dFEV1 and ECP.

**Table 1 children-09-00920-t001:** Baseline demographic and clinical characteristics of the subjects.

	BA	Others	Total
Numbers	51	21	72
Sex (male)	37	10	47 (65.3%)
Age (years)	8.78 ± 2.72	9.90 ± 3.12	9.11 ± 2.87
Height (cm)	133.51 ± 15.39	138.84 ± 16.07	135.07 ± 15.67
Weight (kg)	35.79 ± 15.26	39.92 ± 14.40	36.99 ± 15.03
Cough	36	12	48 (66.7%)
Dyspnea	22	7	29 (40.3%)
Cough on exercise	8	3	11 (15.3%)
Dyspnea on exercise	10	5	15 (20.8%)
Symptoms to cold air	19	6	25 (34.7%)
History of rhinitis	36	11	47 (65.3%)
History of atopic dermatitis	16	9	25 (34.7%)
History of food allergy	14	4	18 (25%)
Exposure to smoking	12	3	15 (20.8%)
Family history of asthma	3	1	4 (5.6%)
Family history of allergy	24	6	30 (41.7%)
Atopy	43	15	58 (81.7%)

Abbreviation: BA—bronchial asthma.

**Table 2 children-09-00920-t002:** Laboratory results of each group.

	BA	Others	*p*
Numbers	51	21	
Sputum eosinophil (%)	5.87 ± 12.1	1.27 ± 1.03	0.014
Eosinophils (/mm^3^)	511.02 ± 361.89	331.25 ± 426.17	0.143
ECP (μg/L)	44.39 ± 40.17	19.58 ± 15.15	0.001
Total IgE (IU/mL)	684.03 ± 841.04	438.91 ± 651.82	0.256
Vitamin D (ng/mL)	25.64 ± 7.17	27.38 ± 12.01	0.872
Numbers of positive skin prick test	5.7 ± 4.54	4.58 ± 3.75	0.302
FVC (%)	104.53 ± 13.88	101.29 ± 10.15	0.336
FEV1 (%)	94.20 ± 17.17	97.14 ± 10.38	0.467
FEV1/FVC	78.24 ± 7.02	84.43 ± 3.28	<0.001
FEF25-75 (%)	81.06 ± 29.24	94.67 ± 22.39	0.060
PC20 (mg/mL)	2.94 ± 3.80	19.29 ± 2.08	<0.001

Values are presented as mean ± standard deviation or as median (minimum–maximum). Abbreviations: BA—bronchial asthma; ECP—eosinophil cationic protein; IgE—immunoglobulin E; FVC—forced vital capacity; FEV1—forced expiratory volume in 1 s; FEF25-75—forced expiratory flow at 25–75% of FVC; PC20—provocation concentration of methacholine causing a 20% fall in FEV1.

**Table 3 children-09-00920-t003:** Correlations between CDA results and other factors in all subjects.

	*p*
Age	0.885
Height	0.835
Weight	0.829
BMI	0.607
Cough	0.504
Dyspnea	0.530
Dyspnea on exercise	0.961
Symptoms to cold air	0.622
History of atopic dermatitis	0.521
History of food allergy	0.633
Exposure to smoking	0.744
Family history of asthma	0.936
Family history of allergy	0.812
Atopy	0.521
Sputum eosinophil	0.020
Eosinophils	0.353
ECP	0.085
Total IgE	0.890
Positivity of specific IgE	0.587
Positivity of skin prick test	0.525
Positive numbers in skin prick test	0.650
FVC	0.204
FEV1	0.168
FEV1/FVC	0.735
FEF25-75	0.447
PC20	0.188

Abbreviations: CDA—cold dry air; BMI—body mass index; ECP—eosinophil cationic protein; IgE—immunoglobulin E; FVC—forced vital capacity; FEV1—forced expiratory volume in 1 s; FEF25-75—forced expiratory flow at 25–75% of FVC; PC20—provocation concentration of methacholine causing a 20% fall in FEV1.

**Table 4 children-09-00920-t004:** Comparisons between CDA positive and CDA negative subjects in asthma patients.

	CDA Positive	CDA Negative	*p*
Eosinophilic BA	2	12	
Non-eosinophilic BA	25	12	0.001
Classic BA	21	16	
Cough-variant BA	6	8	0.375
Cough	21	15	0.232
Dyspnea	12	10	0.842
Cough on exercise	2	6	0.085
Dyspnea on exercise	3	7	0.105
Symptoms to cold air	10	9	0.686
Sputum eosinophil (%)	1.6	10.1	0.019
Eosinophils (/mm^3^)	457.2	567.1	0.293
ECP (μg/L)	34.3	57.4	0.047
Total IgE (IU/mL)	26.0	25.3	0.704

Abbreviations: CDA—cold dry air; BA—bronchial asthma; ECP—eosinophil cationic protein.

**Table 5 children-09-00920-t005:** Comparisons between BA with rhinitis group and BA only group.

	BA with Rhinitis	BA Only	*p*
Numbers	33	18	
Age (years)	9.24 ± 3.06	7.94 ± 1.73	0.059
Height (cm)	137.22 ± 16.68	126.73 ± 9.87	0.018
Weight (kg)	38.45 ± 17.39	30.92 ± 8.73	0.045
Sputum eosinophil (%)	5.90 ± 10.73	5.81 ± 14.70	0.982
Eosinophils (/mm^3^)	517.19 ± 342.74	499.41 ± 406.30	0.872
ECP (μg/L)	40.54 ± 40.42	52.09 ± 39.81	0.353
Total IgE (IU/mL)	666.01 ± 795.29	716.07 ± 940.09	0.842
Numbers of positive skin prick test	5.53 ± 4.03	6.0 ± 5.44	0.730
FVC (%)	102.94 ± 15.82	107.44 ± 9.01	0.272
FEV1 (%)	92.18 ± 19.51	97.89 ± 11.35	0.261
FEV1/FVC	77.53 ± 6.77	79.54 ± 7.45	0.331
FEF25-75 (%)	76.18 ± 27.65	90.0 ± 30.74	0.107
PC20 (mg/mL)	2.50 ± 3.23	3.74 ± 4.68	0.302
Positive CDA	19 (57.6%)	8 (44.4%)	0.379
PT10 (min)	8.34 ± 3.36	8.54 ± 2.12	0.916

Values are presented as mean ± standard deviation or as median (minimum–maximum). Abbreviations: BA—bronchial asthma; ECP—eosinophil cationic protein; IgE—immunoglobulin E; FVC—forced vital capacity; FEV1—forced expiratory volume in 1 s; FEF25-75—forced expiratory flow at 25–75% of FVC; PC20—provocation concentration of methacholine causing a 20% fall in FEV1; CDA—cold dry air; PT10—provocation time causing a 10% decrease in FEV1.

## Data Availability

The data presented in this study are available on request from the corresponding author. The data are not publicly available due to ethics.

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
