# Peer review of "Application of a Cold Dry Air Provocation Test in Pediatric Patients with Asthma"

_children, 2022, doi:10.3390/children9060920_

Round 1

Reviewer 1 Report

Ahn et al. accessed the possible clinical application of a cold, dry air provocation test in asthmatic children with and without concomitant allergic rhinitis.

The topic is interesting.

Abstract – please clarify the abbreviation, especially “PT10”, and the number of asthmatic children.

The introduction is informative and appropriate.

Materials and methods – please specify if the “rhinitis group” include all children with rhinitis – allergic and non-allergic.

Are the asthmatic children on control therapy, specifically ICS, at the moment of enrolment and CDA assessment? If yes, please specify.

The methods are well described.

In Results for better understanding – the “all subjects” or the main group could be defined as “children with suspected asthma and or respiratory allergies”. It must be highlighted that the selected non-asthmatic group children are not representative of the general population and could not serve as a specific control group.  

Lines 194-196 Please rephrase the sentence structures like: “The decrease in CDA15 dFEV1 was greater as PC20 decreased (r=0.336, 194 P=0.017), and PT10 was shorter as FEV1/FVC (r=0.547, P=0.043) and FEF25-75 (r=0.534, 195 P=0.049) were less in all subjects.” 

 And lines 205-211 “In all asthma patients, the decrease in CDA5 205 dFEV1 was greater as PC20 decreased (r=0.295, P=0.049) and total IgE increased (r=- 206 0.302, P=0.033). The decrease in CDA15 dFEV1 was greater as PC20 (r=0.36, P=0.015), 207 sputum eosinophil (%) (r=0.305, P=0.039), and ECP was less (r=0.296, P=0.041) (Fig. 2). 208 PT10 was shorter as the eosinophil count (r=0.731, P=0.011), FVC (r=0.636, P=0.019), FEV1 209 (r=0.611, P=0.027), FEV1/FVC (r=0.599, P=0.031), and FEF25-75 (r=0.615, P=0.025) were 210 less and shorter as the BMI increased (r=-0.608, P=0.027). “ in order to be more understandable.

The discussion is relevant. Please add more recent studies in the references (only three references are since 2015). The text needs English polishing, especially the conclusion.

Author Response

1. Abstract – please clarify the abbreviation, especially “PT10”, and the number of asthmatic children.

Response: Thank you for your comment.

- We have defined “PT10” for clarity; PT10 was the provocation time causing a 10% decrease in FEV1

Among the 72 subjects studied in the present study, 51 were diagnosed with asthma.

2. Materials and methods – please specify if the “rhinitis group” include all children with rhinitis – allergic and non-allergic.

 Response: Thank you for your comment.

- Asthma with rhinitis group includes all children with allergic and non-allergic rhinitis.

We have added the following sentence to the last paragraph of the Introduction: “Among asthma patients, those with allergic and/or non-allergic rhinitis.” 

3. Are the asthmatic children on control therapy, specifically ICS, at the moment of enrolment and CDA assessment? If yes, please specify.

 Response: Thank you for your comment.

- The enrolled subjects had not used some medications for asthma, including ICS. They were diagnosed with asthma after visiting our clinic and undergoing tests. In addition, they were not on ICS therapy at enrollment and CDA assessment.

4. In Results for better understandingthe “all subjects” or the main group could be defined as “children with suspected asthma and or respiratory allergies”. It must be highlighted that the selected non-asthmatic group children are not representative of the general population and could not serve as a specific control group.  

 Response: Thank you for your comment.

- We have changed “3.1. CDA Provocation Tests in All Subjects” to “3.1. CDA Provocation Tests in the Children with Suspected Asthma and/or Respiratory Allergies

5. Lines 194-196 Please rephrase the sentence structures like: “The decrease in CDA15 dFEV1 was greater as PC20 decreased (r=0.336, 194 P=0.017), and PT10 was shorter as FEV1/FVC (r=0.547, P=0.043) and FEF25-75 (r=0.534, 195 P=0.049) were less in all subjects.” 

 Response: Thank you for your comment.

- We have rephrased the sentence as follows: CDA15 dFEV1 correlated with PC20 (r=0.336, P=0.017), and PT10 became shorter as FEV1/FVC (r=0.547, P=0.043) and FEF25-75 (r=0.534, P=0.049) decreased in all subjects

6. And lines 205-211 “In all asthma patients, the decrease in CDA5 dFEV1 was greater as PC20 decreased (r=0.295, P=0.049) and total IgE increased (r=- 0.302, P=0.033). The decrease in CDA15 dFEV1 was greater as PC20 (r=0.36, P=0.015), sputum eosinophil (%) (r=0.305, P=0.039), and ECP was less (r=0.296, P=0.041) (Fig. 2). PT10 was shorter as the eosinophil count (r=0.731, P=0.011), FVC (r=0.636, P=0.019), FEV1 (r=0.611, P=0.027), FEV1/FVC (r=0.599, P=0.031), and FEF25-75 (r=0.615, P=0.025) were less and shorter as the BMI increased (r=-0.608, P=0.027). “ in order to be more understandable.

Response: Thank you for your comment.

- We have rephrased the sentences as follows: In all asthma patients, CDA5 dFEV1 correlated with PC20 (r=0.295, P=0.049) and total IgE (r=- 0.302, P=0.033). CDA15 dFEV1 correlated with PC20 (r=0.36, P=0.015), sputum eosinophil (%) (r=0.305, P=0.039), and ECP (r=0.296, P=0.041) (Fig. 2). PT10 became shorter as the eosinophil count (r=0.731, P=0.011), FVC (r=0.636, P=0.019), FEV1 (r=0.611, P=0.027), FEV1/FVC (r=0.599, P=0.031), and FEF25-75 (r=0.615, P=0.025) decreased and shorter as the BMI increased (r=-0.608, P=0.027).

7. The discussion is relevant. Please add more recent studies in the references (only three references are since 2015).

Response: Thank you for your comment.

- We have added more recent references.

8. The text needs English polishing, especially the conclusion.

 Response: Thank you for your comment.

- This manuscript has been proofread for English language by a native English speaker. We have attached a certificate of English proofreading.

Reviewer 2 Report

The study is interesting and properly conducted. Numerous parameters were measured, both as regards the allergological and functional components, as well as trying to endotype the asthma of young patients as best as possible. The idea of ​​identifying an alternative indirect broncho-costriction test (as test with mannitol for example) for exertional asthma, which not so infrequently have normal lung volumes and negative methacholine tests, is undoubtedly appreciable. The results of the study would seem to have clinical, practical, very attractive implications. The CAD is relatively fast and feasible, no particular adverse events have emerged, or at least the authors have not explicitly mentioned them. The study, as indeed, written by the authors has strong limitations: first of all the low number of subjects, second that there is no significant correlation between test positivity and exertional dyspnea. Even the hypothesis of grading the severity of asthma based on the PT10 dose results “a bit risky” given the small numbers.

The CAD test requires validation and standardization, drastically increasing the number of cases

It is therefore suggested that the methods of discussing the data be generally attenuated and, above all considering  the objective limits of the study, the conclusions.

Author Response

The study is interesting and properly conducted. Numerous parameters were measured, both as regards the allergological and functional components, as well as trying to endotype the asthma of young patients as best as possible. The idea of ​​identifying an alternative indirect broncho-costriction test (as test with mannitol for example) for exertional asthma, which not so infrequently have normal lung volumes and negative methacholine tests, is undoubtedly appreciable. The results of the study would seem to have clinical, practical, very attractive implications. The CAD is relatively fast and feasible, no particular adverse events have emerged, or at least the authors have not explicitly mentioned them. The study, as indeed, written by the authors has strong limitations: first of all the low number of subjects, second that there is no significant correlation between test positivity and exertional dyspnea. Even the hypothesis of grading the severity of asthma based on the PT10 dose results “a bit risky” given the small numbers. The CAD test requires validation and standardization, drastically increasing the number of cases. It is therefore suggested that the methods of discussing the data be generally attenuated and, above all considering the objective limits of the study, the conclusions.

Response: Thank you for your comment.

- We have added the following text in the beginning of the discussion: “CDA provocation test was relatively fast and feasible, and no particular adverse events emerged.”

 - The small number of participants is one of the major limitations of our study. This is a single-center study, and we enrolled the subjects who visited our clinic and had not been diagnosed with asthma by clinicians. In addition, we enrolled the subjects who performed all the tests, including allergy test, pulmonary function test, methacholine provocation test, and CDA provocation test. Notably, a 24-hour interval is required between methacholine and CDA provocation tests. Thus, the number of participants is quite small. Additional large-scale studies are needed, and we have described this in the Conclusion.

 - Also, some patients had missing medical records related to symptoms during normal exposure to cold air and dyspnea on exercise. Thus, the evaluation of the correlation between symptoms and CDA provocation might not have been sufficient. We have added this shortcoming to the Limitations paragraph.

 - About PT10, we have noted that there is a possibility that the PT10 of the CDA provocation test could be used as an indicator of asthma severity. We have also mentioned that additional large-scale studies compensating for the limitations are needed.

Reviewer 3 Report

The paper "Application of a Cold Dry Air Provocation Test in Pediatric Patients with Asthma" by Ji Young Ahn et al. is an analysis of a alternative or additional test in children with asthma. There are some interesting results but also some important issues that must be discussed. 

page 2, line 59. 72 children in about 16 months. The number is quit small. The prevalence of asthma in children in higher than in adults and increasing. 

Page 2, line 87. Only 6 allergens were tested? 

Page 4, table 2. FEF 25-75% was not significantly different. Only the FEV1/FVS. How do you interpret this? 

Page 5, chapter 3.1. Is there a table with the results? 

Page 9, line 278. Additional large-scale studies are needed! That is the most important conclusion!

Page 10, references. Most of the papers are old, very few are recent. The cold air provocation test is not research intensively? Maybe the test we have are sufficient? 

Do you have any data about FENO? 

Author Response

1. page 2, line 59. 72 children in about 16 months. The number is quite small. The prevalence of asthma in children in higher than in adults and increasing. 

Response: Thank you for your comment.

- We have acknowledged that the small sample size is one of the major limitations of our study. This is a single-center study, and we enrolled the subjects who visited our clinic and had not been diagnosed with asthma by clinicians. In addition, we enrolled the subjects who performed all the tests, including allergy test, pulmonary function test, methacholine provocation test, and CDA provocation test. Notably, a 24-hour interval is needed between methacholine and CDA provocation tests. Thus, the number of participants is quite small. Additional large-scale studies are needed, and we have described this in the Conclusion.

2. Page 2, line 87. Only 6 allergens were tested? 

Response: Thank you for your comment.

- In South Korea, because of insurance coverage limitation, only 6 allergens can be tested using the CAP radioallergosorbent technique. Thus, we performed the skin prick test for 18 allergens simultaneously.

3. Page 4, table 2. FEF 25-75% was not significantly different. Only the FEV1/FVC. How do you interpret this? 

Response: Thank you for your comment.

- The FEF25–75% values between BA and Others were different, but not significantly. A FEF25–75% is considered an early marker of bronchial hyperresponsiveness and small airway inflammation. In our study, others group included subjects who were not diagnosed with asthma using FEV1/FVC and/or PC20; those in early stages of bronchial hyperresponsiveness might also be included in this group. Also, patients with allergic rhinitis were included in the others group. These are the potential reasons why the FEF25–75% values between BA and Others were not significantly different.

4. Page 5, chapter 3.1. Is there a table with the results? 

Response: Thank you for your comment.

- We added new table 3 with the results of chapter 3.1.

5. Page 9, line 278. Additional large-scale studies are needed! That is the most important conclusion!

Response: Thank you for your comment.

- We have added, “Additional large-scale studies compensating for the limitations are needed” in the Conclusion.

6. Page 10, references. Most of the papers are old, very few are recent. The cold air provocation test is not research intensively? Maybe the test we have are sufficient? 

Response: Thank you for your comment.

- Cold air generators that produce dry air at below-freezing temperatures are commercially available and are in use in some laboratories.

The ERS Official Documents published in 2018 identified cold air challenge. In this Document, cold air inhalation enhances the response to eucapnic voluntary hyperpnoea in children with asthma. The addition of cold air appears to shorten the stimulus time necessary during the challenge test from 6 to 4 min. In patients with symptoms specifically associated with exercising in the cold, exercise challenge while breathing cold dry air may be useful to enhance the sensitivity and specificity of the response. Notably, part of the response to exercising in cold temperatures may be due to exposure of the face and body to cold temperatures, and not just the airways. Using cold air as part of an exercise or eucapnic voluntary hyperpnoea challenge elicits responses that are discordant with direct challenge tests, such as methacholine or histamine.

We have added more recent references.

Do you have any data about FENO? 

We have been collecting new and follow-up data about FeNO. We are conducting the FeNO test when asthma patients perform follow-up pulmonary function test and methacholine provocation test. Also, we are performing the FeNO test when patients with chronic cough, shortness of breath, and wheezing visit our clinic for asthma diagnosis. However, we have not analyzed the FeNO data yet. Once we collect enough FeNO data, we will perform additional analysis.

Round 2

Reviewer 1 Report

The authors addressed all the issues and recommendations. As a result, the new version has improved quality. Unfortunately, the track-changes mode was not used. The last is essential in the reviewing process. 

The authors have added new references; however, the discussion could be further improved by adding some recent points of view on provocation testing in children. 

There are some suggestions for adding to the discussion:

http://dx.doi.org/10.1016/j.rmed.2012.06.017

https://doi.org/10.4168/aair.2018.10.1.43

10.3389/fped.2021.773794